# CT characteristics and diagnostic value of COVID-19 in pregnancy

**XiaoMing Gong**[1,2], **Lu Song**[1], **Hang Li**[3], **Li Li**[4], **Wei Jin**[5], **KaiHu Yu**[2], **Xiaochun Zhang**[1], **Hongjun Li**[4], **HengNing Ke**[6]*, **ZhiYan Lu**[1]*

**1** Department of Radiology, Zhongnan Hospital of Wuhan University, Wuhan, China, **2** Department of Radiology, Xianning central Hospital, Xianning, China, **3** Department of Radiology, Hospital of Stomatology Wuhan University, Wuhan, China, **4** Department of Radiology, Beijing Youan Hospital, Capital Medical Universtiy, Beijing, China, **5** School of Public Health, Indiana University, Indianapolis, Indiana, United States of America, **6** Department of Infectious Diseases, Training Center of AIDS prevention and Cure of Hubei Province, Zhongnan Hospital of Wuhan University, Wuhan, China

* 15729577635@163.com (HNK); luzy100@163.com (ZYL)

**Data Availability Statement:** All relevant data are within the paper.

**Funding:** We declare that National Nature Project (81572902), Chinese Center for Disease Control and Disinfection Special Fund(201865), Hubei

## Abstract

### Objective

To investigate the computed tomography (CT) characteristics and diagnostic value of novel coronavirus pneumonia (NCP or COVID-19) in pregnancy.

### Methods

This study included ten pregnant women infected with COVID-19, treated in the Zhongnan Hospital of Wuhan University from January 20, 2020 to February 6, 2020. Clinical and chest CT data were collected and clinical symptoms, laboratory indicators, and CT images were analyzed to explore CT characteristics and diagnostic value for COVID-19 during pregnancy.

### Results

Laboratory examination showed that white blood cell count was normal in nine patients, and slightly higher in one patient (10.23 × 109). The lymphocyte ratio decreased in two patients by 12% and 14%, respectively. The levels of C-reactive protein was elevated in seven patients (range, 21.16–60.3 mg/L) and the levels of D-dimer was increased in eight patients (range, 507–2141 ng/mL). Six patients had low levels of total protein (range, 35.3–56.5 mg/L). Two patients showed small patchy ground glass opacity (GGO) involving single lung, while eight patients showed multilobe GGO in both the lungs, with partial consolidation. Peripheral and non-peripheral lesion distributions were seen in ten (100%) and four (40%) patients, respectively. There were four patients who had signs of intra-bronchial air-bronchogram, six patients had small bilateral pleural effusions, while none had lymphadenopathy. Dynamic observations were performed in four patients after COVID-19 treatment. Among these four patients, one patient showed normal on the initial examination, and new lesions were observed after 3 days; 1 patient showed progression after 7 days of treatment, with expansion of the lesion area; and the other 2 patients showed improvement after 14 days of

Provincial Health and Health Commission Project (Wj2017M044); Science and Technology Innovation Cultivation Fund of Zhongnan Hospital of Wuhan University (znpy2018033) altogether Play an important role in the following function (study design, data collection and analysis, decision to publish, and preparation of the manuscript.).

**Competing interests:** No authors have competing interests.

treatment, with reduction in the density and area of lesions and appearance of linear opacity.

## Conclusions

The CT characteristics of COVID-19 in pregnancy were mainly observed in early and progressive stages, and multiple new lesions were common. And there were consolidations of varying sizes and degrees within the lesion. Moreover, the original ground glass lesions could be fused or partially absorbed. Six patients had small bilateral pleural effusion. In summary, CT scans can play an important role in early screening, dynamic observation, and efficacy evaluation of suspected or confirmed cases of pregnant women with COVID-19.

## Introduction

Since December 2019, a large number of novel coronavirus (2019-nCoV) pneumonia cases have been reported in Wuhan, the capital of Hubei Province and a large city of approximately 11 million persons, located in the central region of the People's Republic of China [1]. This newly recognized β-coronavirus causes COVID-19, which has rapidly spread throughout China and has crossed international borders, owing to human-to-human transmission of the virus via intercontinental travel [2]. As of 24:00, February 4, 2020, a total of 24324 COVID-19 cases in China have been confirmed [3]. During pregnancy, the mother's body undergoes a variety of changes, which include changes in anatomy, bodily functions, and immune status; thus resulting in an immunosuppressive state. The newly discovered 2019-nCoV is a large number of people are susceptible to the newly discovered severe acute respiratory syndrome coronavirus 2 (SARS-CoV-2). [1]. It is known that 2019-nCoV can infect pregnant women; such women are more susceptible to COVID-19, and the disease can cause potential maternal and fetal complications [3]. Therefore, increased attention should be given to COVID-19 patients who are pregnant. Unfortunately, there is limited experience with COVID-19 during pregnancy. In addition, there have been no reports on the imaging manifestations of COVID-19 during pregnancy, in China or other countries. According to "Diagnosis and Treatment of Pneumonia for Novel Coronavirus Infection" (Trial Version 5) of the National Health Commission, images of 2019-nCoV pneumonia showed rare pleural effusion [1]. However, COVID-19 in pregnancy is often found to be associated with pleural effusion. In this study, a retrospective analysis was conducted using the clinical data and computerized tomographic (CT) images of the chest of pregnant women with COVID-19, treated in our hospital from January 20, 2020 to February 6, 2020. The generated data was summarized to improve the understanding and diagnosis of COVID-19 in pregnancy.

## Materials and methods

### Study design and patients

We did a retrospective review of medical records of ten pregnant women with COVID-19, admitted to the Zhongnan Hospital of Wuhan University from January 20, 2020 to February 6, 2020. Diagnosis of COVID-19 pneumonia was based on the New Coronavirus Pneumonia Prevention and Control Program (5th edition), published by the National Health Commission of China. Six pregnant women with COVID-19 pneumonia tested positive for severe acute respiratory syndrome coronavirus 2 (SARS-CoV-2), based on the quantitative reverse

transcription polymerase chain reaction (qRT-PCR) analysis of the samples from the respiratory tract. The other four cases were clinically diagnosed according to Diagnosis and Treatment of Pneumonia for Novel Coronavirus Infection (Trial Version 5). This study was reviewed and approved by the Medical Ethical Committee of Zhongnan Hospital of Wuhan University (approval number 2020037). Written informed consent was obtained from each enrolled patient.

Maternal throat swab samples were collected and tested for SARS-CoV-2 using the Chinese Center for Disease Control and Prevention (CDC) recommended kit (BioGerm, Shanghai, China), following WHO guidelines for qRT-PCR. [4]. All samples were processed simultaneously at the Department of Clinical Laboratory of Zhongnan Hospital and State Key Laboratory of Virology/Institute of Medical Virology, School of Basic Medical Sciences, Wuhan University. Positive cases of COVID-19 infection were defined as those with a positive test result from either laboratory.

The average age of the patients included in this study was 30 years (range, 26 to 40 years). Gestational age ranged from $36^{+1}$ to $39^{+4}$ weeks, with a median age of $37^{+3}$ weeks. The median onset time ranged from 1 to 10 days, with 6 days on median. The symptoms included fever (n = 8), with the body temperature ranging from 37.5 ˚C to 38.3 ˚C; dry cough (n = 5); nasal congestion (n = 1); and paroxysmal abdominal distension and diarrhea (n = 1). Of the two patients who did not have fever, one had cough with little mucus and sputum, and the other was hospitalized due to abdominal distention and diarrhea. Pregnancy complications were reported in two patients, one with gestational diabetes mellitus and the other with hypothyroidism. Laboratory examination showed that white blood cell count was normal in nine patients, and slightly higher in one patient ($10.23 \times 10^9$). The lymphocyte ratio was decreased by 12% and 14% in two patients. The levels of C-reactive protein (CRP) was elevated in seven patients (range, 21.16–60.3 mg/L), while the levels of D-dimer were increased in eight patients (range, 507–2141 ng/mL). The total protein level was low in six patients (range, 35.3–56.5 mg/L). Hypersensitive troponin was increased in one patient (77.8 pg/mL). A summary of the various signs and symptoms of the ten pregnant women with COVID-19 are shown in Table 1.

**Table 1. Demographic signs and symptoms of 10 pregnant women with COVID-19.**

| Numbering | Age (y) | First symptom | Maximum body temperature (˚C) | Laboratory indicators | | | | | | | | | Disease onset to CT Features (d) | |
|---|---|---|---|---|---|---|---|---|---|---|---|---|---|---|
| | | | | RBC (×10^12/L) | Hb(g/L) | WBC (10^9/L) | Lymphocyte percentage(%) | Neutrophil percentage(%) | C-reactive protein (mg/L) | D-D dimer (ng/ml) | Total protein (g/L) | First CT | CT review |
| 1 | 27 | Cough | 37.6 | 4.25 | 113.4 | 7.55 | 14.0 | 82.0 | 8.80 | 2080 | 75.7 | 7 | |
| 2 | 27 | Cough | 38.3 | 3.66 | 111.0 | 5.07 | 22.3 | 73.5 | 9.50 | 2141 | 56.0 | 2 | |
| 3 | 27 | Fever | 38.3 | 4.07 | 108.7 | 7.22 | 21.7 | 76.1 | 17.85 | 1132 | 49.2 | 3 | |
| 4 | 33 | Normal | 36.5 | 4.12 | 133.7 | 6.15 | 12.0 | 73.4 | 60.30 | 1251 | 56.5 | 3 | 6 |
| 5 | 26 | Nasal congestion | 37.8 | 4.04 | 127.0 | 9.34 | 22.5 | 74.2 | 6.99 | 507 | 35.3 | 2 | 5 |
| 6 | 34 | Cough | 37.5 | 4.17 | 123.0 | 7.08 | 21.5 | 73.6 | 24.87 | 1106 | 66.8 | 7 | |
| 7 | 26 | Diarrhea | 37.5 | 3.57 | 159.0 | 8.29 | 22.5 | 75.3 | 9.50 | 377 | 52.9 | 1 | 4 |
| 8 | 40 | bellyache | 36.5 | 3.20 | 93.9 | 10.23 | 21.0 | 87.2 | 34.90 | 412 | 48.7 | 3 | |
| 9 | 27 | Cough | 37.6 | 4.39 | 134.0 | 6.98 | 21.9 | 72.7 | 18.40 | 1090 | 56.4 | 2 | |
| 10 | 27 | Cough | 37.9 | 3.98 | 105.0 | 6.59 | 18.3 | 79.2 | 57.22 | 920 | 35.6 | 8 | 12 |

## Examination method

CT scans were performed using Siemens 64 and Philips 64 CT scanners. The CT protocol used was as follows: tube voltage = 120kV; automatic tube current; pitch = 1.0; section thickness = 0.625 mm; interval = 5.0 mm; axial reconstruction thickness = 0.625 mm; matrix $512 \times 512$. The following windows were used: a lung window with a window width of 1000 HU and a window level of -600 HU, and a mediastinal window with a window width of 300 HU and a window level of 40 HU.

Two senior radiologists reviewed the images independently, mainly observing the distribution, morphology, density, and dynamic changes in lesions, including pleural effusion and mediastinal lymph node. In case of discrepancy, a conclusion was reached by consensus. All patients signed informed consent before CT examination. For all pregnant women included in this study, the abdomen was protected with lead, and the CT dose was 411 mGY.

Lymphadenopathy was defined as a lymph node >1 cm in short-axis diameter.

## Results

The CT findings of the patients included in this study are shown in Table 2.

Distribution analysis showed the presence of lesions in one lung of two patients (20%), one right and one left lung. Lesions in both the lungs were observed in eight patients (80%). There were seven patients (70%) who had lesions distributed in the upper lobes, nine patients (90%) with lesions distributed in the middle lobe (lingular segment), and ten patients (100%) with lesions distributed in the lower lobe. Ten patients (100%) had lesions distributed in the peripheral region of the lungs and four patients (40%) had lesions distributed in the non-peripheral region.

Morphology analysis showed that there were six patients (60%) with patchy shadow and nine patients (90%) with small patchy shadow.

Density analysis showed that there were ten (100%) patients with pure ground glass opacity (GGO), amongst which six (60%) had GGO accompanied by consolidation (Fig 1), one (10%) had GGO accompanied by reticular and/or interlobular septal thickening, and 4 (40%) showed signs of intra-bronchial air-bronchogram (Fig 2) Six patients (60%) had small bilateral pleural effusions (Fig 3), while no patient had lymphadenopathy. Dynamic changes and prognosis of

**Table 2. CT findings of 10 cases of pregnancy with COVID-19.**

| Findings | case1(%) | case2 | case3 | case4 | case5 | case6 | case7 | case8 | case9 | case10 | Number of patients |
|---|---|---|---|---|---|---|---|---|---|---|---|
| Unilateral lung | – | – | – | + | – | + | – | – | – | – | 2(20%) |
| Bilateral lung | + | + | + | – | + | – | + | + | + | + | 8(80%) |
| Upper lobes | + | + | + | – | – | – | + | + | + | + | 7(70%) |
| Middle lobe | + | + | + | – | + | + | + | + | + | + | 9(90%) |
| Lower lobes | + | + | + | + | + | – | + | + | + | + | 10(100%) |
| Flake | + | + | + | – | – | – | – | + | + | + | 6(60%) |
| Patchy | + | + | + | – | + | + | + | + | + | + | 9(90%) |
| Pure GGO | + | + | + | + | + | + | + | + | + | + | 10(100%) |
| Consolidation | + | + | + | + | – | – | – | – | + | + | 6(60%) |
| Reticulation | + | – | – | – | – | – | – | – | – | – | 1(10%) |
| Fiber rope | – | – | + | – | – | – | + | + | – | – | 3(30%) |
| Air bronchogram | + | + | + | – | – | + | – | – | – | – | 4(40%) |
| Pleural effusion | – | + | + | + | – | + | + | – | – | + | 6(60%) |
| Lymphadenopathy | – | – | – | – | – | – | – | – | – | – | 0(0%) |

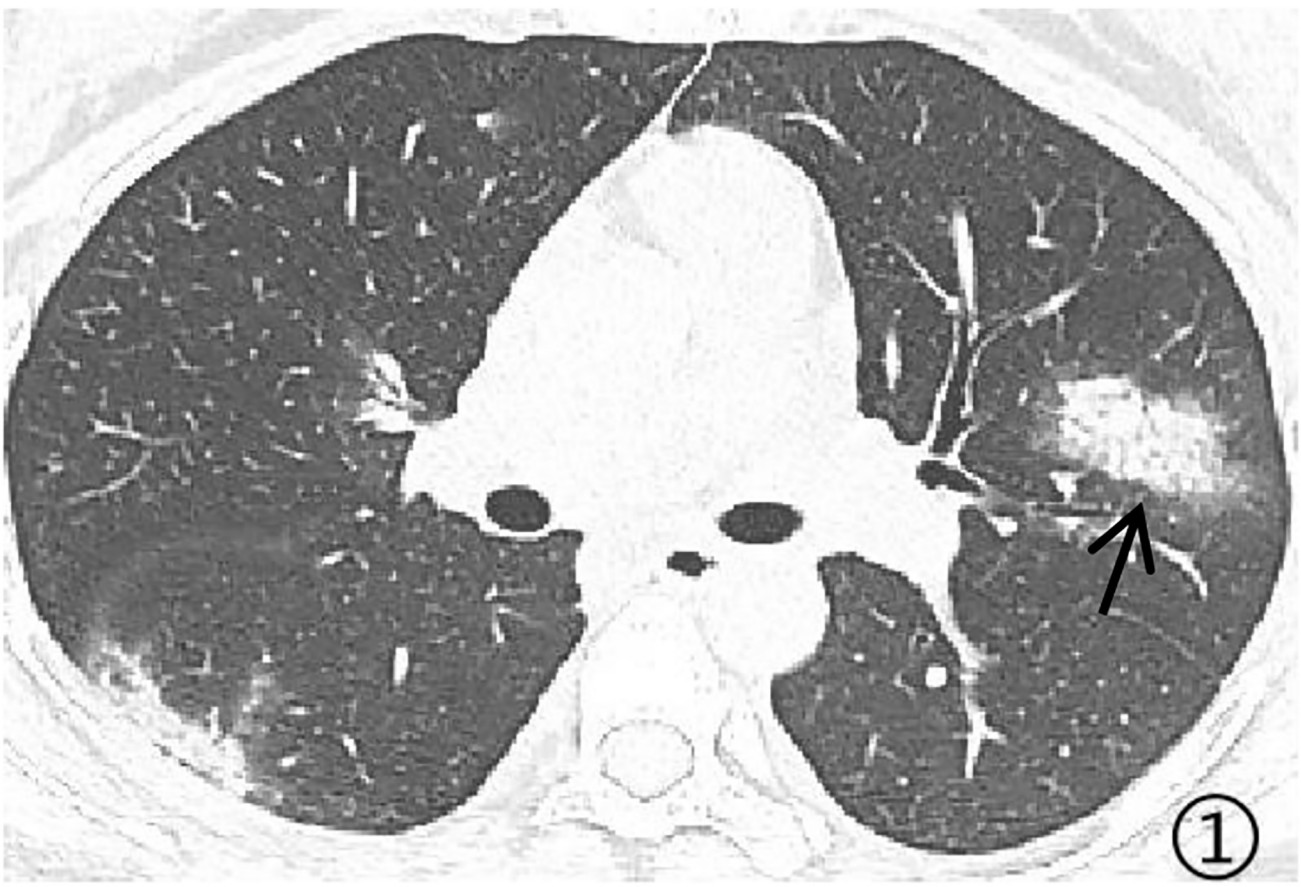

**Fig 1. A 27-year old female with menopause for 38$^{+2}$ weeks presented with fever and cough for 2 days.** Her husband was diagnosed with COVID-19 three days ago. She was diagnosed with COVID-19. The left upper lobe and the dorsal segment of the right lower lobe showed patchy shadow. Faint density shadows were seen throughout the lungs, displaying a halo sign.

lesions were analyzed during the study. Chest CT follow-up examination was performed in four patients. Two patients showed improvement in symptoms, with a lighter density of the lesion and reduction in the area of the lesions (Fig 4a and 4b); 1 patient showed normal on the initial examination, and a small patchy GGO, with signs of intra-bronchial air-bronchogram observed in the left upper lobe after 3 days of observation (Fig 5a and 5b); one patient showed progressive disease after 7 days of observation, with expansion in lesion area, increase in density, and appearance of linear opacity (Fig 6a and 6b). All newborn pharyngeal swabs were tested twice for novel-coronavirus nucleic acid and were found to be negative.

## Discussion

The CT findings of ten pregnant women with COVID-19 were as follows: (1) 60% of patients had small bilateral pleural effusion, which was not in line with the previous reports which indicated that pleural effusion is rare in COVID-19 [5]. Pleural effusion may be associated with either COVID-19 or pregnancy status. This may be attributed to the following reasons: a) the total protein levels in these six patients had decreased to varying degrees, ranging from 35.3 mg to 56.5 mg/L, and a small amount of pleural effusion might have been caused by the decrease in plasma colloid osmotic pressure due to slight hypoproteinemia; b) The CRP levels were elevated in all the six patients, ranging from 21.17.85–60.3 mg/L, thus indicating certain

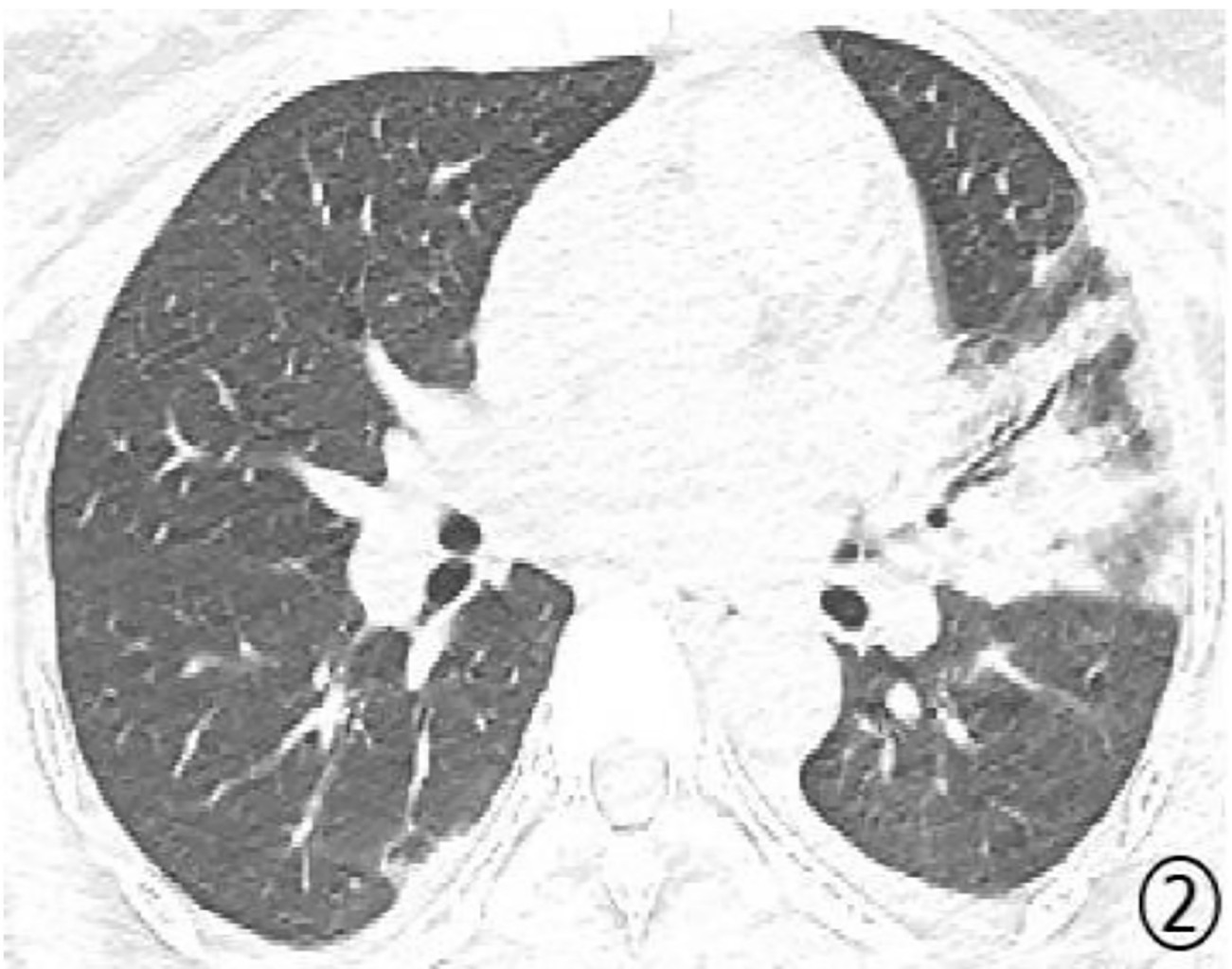

**Fig 2. A 27-year old female with menopause for 38⁺⁶ weeks was confirmed as positive for COVID-19.** Patchy consolidation could be seen in left lingular segment within sign of intra-bronchial air-bronchogram. A small effusion in bilateral pleural cavity was confirmed.

inflammatory reactions. The infection might have caused thickening and congestion of the visceral pleura, and increased vascular permeability, resulting in induction of pleural effusion. The gradient movement of exudate across the visceral pleura in a gradient might have resulted in pleural effusion [6]. c) All the patients in this group were women with late pregnancy at 36–38 weeks of gestation, and late pregnant women are prone to chest leakage due to increased blood volume. (2) All the ten patients included in this study were in the early or progressive stages of COVID-19, and there was no patient in critical stage: six cases (60%) were in early stage based on CT images: two patients (20%) showed single small patchy GGO in one lobe, and four patients (40%) showed multiple patchy GGO in the periphery of both the lungs. Among them, one patient showed normal in the first CT scan, which was consistent with the atypical COVID-19 performance reported by Chung M et al [7]. The other four patients (40%) were in the progressive stage, showing multiple small or large patchy GGO in both the lungs. Some lesions were consolidated with signs of intra-bronchial air-bronchogram. This was consistent with the characteristics of early and mid-stage lesions reported by Song F, et al. [8]. The lesions were commonly distributed in both the lungs with multiple lobes (n = 8), which was

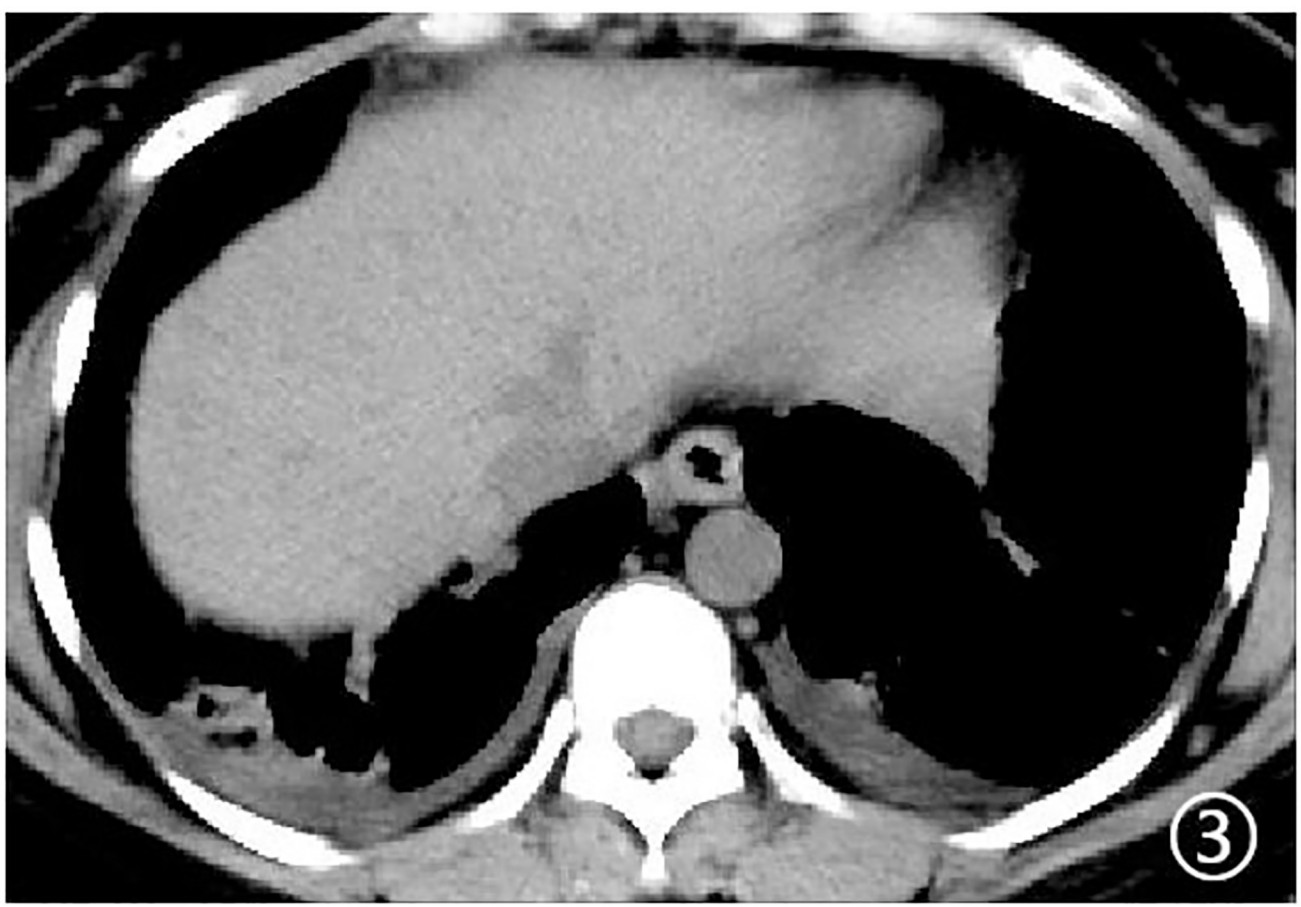

**Fig 3. A 27-year old female with menopause for 37$^{+3}$ weeks.** Mediastinal window showed bilateral pleural effusion.

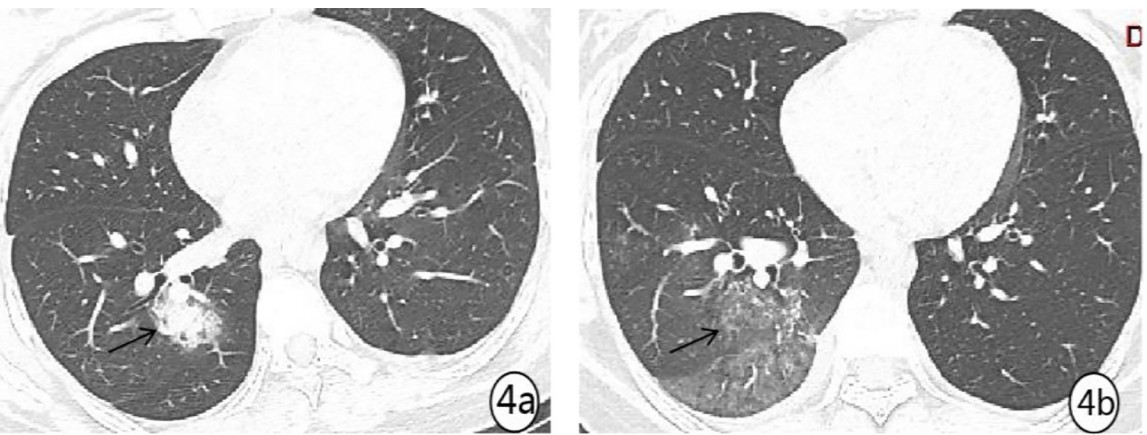

**Fig 4. A 33-year old female with menopause for 37$^{+2}$ weeks, was diagnosed with COVID-19.** (a) chest CT at admission. Patchy increased density was seen in the right lower lung, with bronchiectasis, increased small vascular network, and ground glass opacity (GGO) throughout. (b) same area as 4a after 3 days of treatment. A re-examination showed obvious absorption and thinning density of the lesion, which was replaced by light GGO.

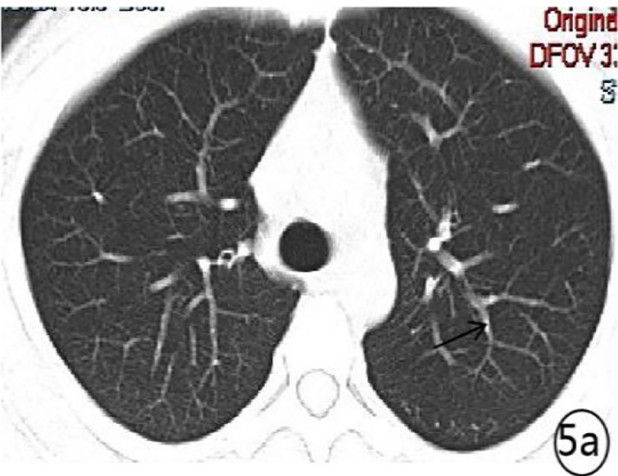
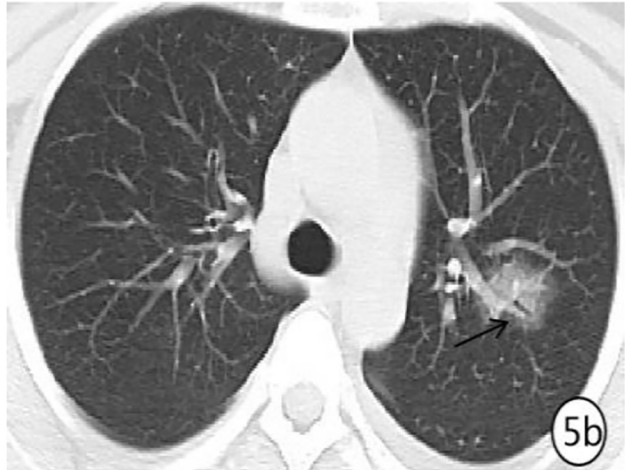

**Fig 5. A 26-year old female with menopause for 36⁺² weeks presented with fever and paroxysmal abdominal distension for 2 days.** She was diagnosed with COVID-19. (a) showed normal on the chest CT at admission. 3 days later, (b) CT re-examination revealed a new lesion in the left lingular segment, showing a patchy GGO with bronchiectasis.

different from bacterial pneumonia and consistent with the report of Shi HS et al. [9]. COVID-19 is difficult to differentiate from SARS, MERS, and other diseases based on CT images. It is thus necessary to combine epidemiology and pathogenic examination for efficient diagnosis of COVID-19 in pregnant women.

Similar to SARS-CoV and MERS-CoV, the SARS-CoV-2 is caused by a β coronavirus. The similarity between the 2019-nCov and SARS-CoV genomes was reported to be as high as 85%. Over the past 20 years, SARS and MERS have caused a total of more than 10,000 patient infections worldwide, with a fatality rate of SARS at 10%, SARS in pregnant women at 25%, and MERS at 37% [3, 10]. According to the latest report, the fatality rate of COVID-19 is 4.3%, [11]. A vast majority of the population is susceptible to SARS-CoV-2 infection, and COVID-

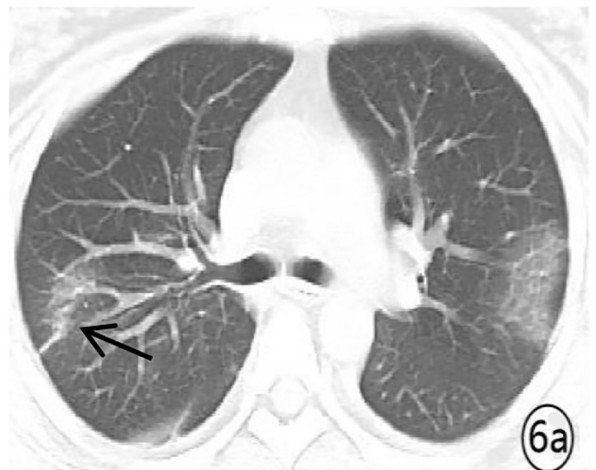
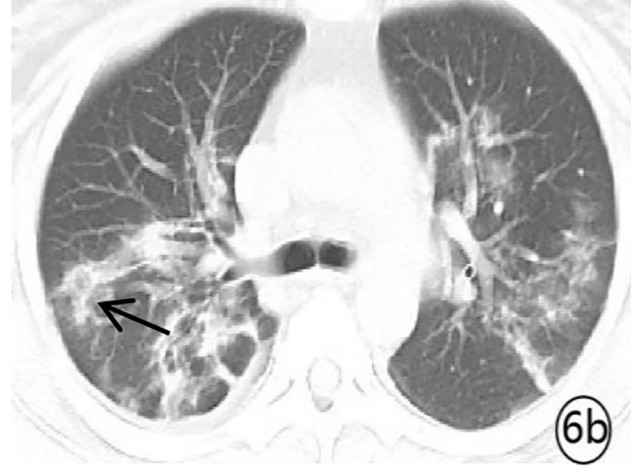

**Fig 6. A 33-year old female with menopause for 37⁺² weeks presented with abdominal distension and diarrhea for 2 days.** She was diagnosed with COVID-19. (a) chest CT at admission showed small flashed-glass shadow in the posterior segment of the upper lobe of the right lung and the lobe of the left tongue. 7 days later. (b) re-examination showed a significant increase in lesions in both the lungs with increased density. It was observed as strip shadow, surrounding ground glass opacity, with interlobular septal thickening.

19 in pregnancy may occur at all gestational ages [12]. The inflammatory stress response in pregnant women to viral pneumonia increases significantly, resulting in rapid development of the disease, especially in the middle and late stages of pregnancy, which are prone to severe illness and may cause maternal-fetal death [13]. At present, the diagnosis of COVID-19 depends on nucleic acid testing, and in clinical practice, there are a few cases of pregnant women with negative 2019-nCoV nucleic acid testing. However, there have been cases of pregnant women who had a history of contact with COVID-19 patients and have typical clinical imaging manifestations of COVID-19. During this time, chest imaging examination (especially CT) has important reference value for the diagnosis and treatment dynamic evaluation of COVID-19 in pregnancy.

In the early stage of embryonic development, high dose exposure of radiation (>1 Gy) can be fatal to the embryo. However, the dose of diagnostic imaging in this study was much lower than 1 G. In addition, radiation exposure at 8 to 15 weeks of gestation has the greatest impact on the central nervous system of the fetus, and some scholars suggest that the lowest dose of exposure that can cause mental retardation of the fetus is above 610 mGy [14,8]. According to the American Radiological Association and the American College of Obstetricians and Gynecologists, the fetal radiation dose in pregnant women undergo a single chest X-ray examination is 0.0005 to 0.01 mGy. In case of chest CT or CT pulmonary angiography (0.1 to 10 mGy), the fetus exposure dose is 0.01 ~ 0.66 mGy [8]. Therefore, for pregnant women suspected of SARS-CoV-2 infection, CT or X-ray scans can be used for chest examination.

After analyzing the diagnosis and treatment process, CT has the following diagnostic value for COVID-19 in pregnancy: 1) cases with strong occult and atypical symptoms can be detected. In this study, a 33-year-old pregnant woman at $36^{+2}$ weeks gestation had abdominal distention and diarrhea as the first symptom, but no fever, dry cough, or other symptoms. Routine outpatient blood examination showed a decrease in lymphocyte ratio and an increase in the level of D-dimer. The patient had a history of contact with COVID-19 patients, and the chest CT showed multiple small patchy of GGO lesions involving both the lungs, indicating viral pneumonia, and the novel coronavirus nucleic acid testing showed positive results. 2) The scope, degree, and therapeutic effect of the lesions can be rapidly evaluated. In this group, four patients underwent follow-up CT re-examination. At follow-up, pulmonary involvement was improved in 2 patients, with smaller lesion size and lighter lesion density. These two patients did not terminate pregnancy. During treatment, disease progression was observed in 2 patients, with an increased size and/or increased consolidation. These patients opted for cesarean to terminate the pregnancy, which improved the maternal and fetal outcomes and avoided maternal and fetal death. 3) The nucleic acid testing false negative patients could be identified, which is conducive to early diagnosis, early quarantine, and early treatment. One patient was admitted to hospital due to fever and cough for 2 days. Her husband was diagnosed with COVID-19 three days ago and had close contact with her. Chest CT showed small patchy GGO in the left lingular segment and the right lower lobe, while the novel coronavirus nucleic acid test was negative, which was in accordance with Diagnosis and Treatment of Pneumonia for Novel Coronavirus Infection (Trial Version 5), and was consistent with the clinical diagnosis. 4) CT examination is quick and only takes a few minutes, while the number of nucleic acid test kits is limited, which is time-consuming and may result in false negatives. 5) Chest CT reexamination is one of the main indicators of discharge. Therefore, CT of COVID-19 in pregnancy is conducive to early control of the source of infection. It plays an important role in early detection, early reporting, monitoring dynamic changes, and detection of complications.

There are a few limitations to our study. First, the group of pregnant women with COVID-19 included in this study were in their third trimester of pregnancy; there is still no understanding of pregnant women with COVID-19 in early and mid-term pregnancy. Further, the

number of confirmed cases is relatively small, and we will further accumulate and summarize data for related research.

In conclusion, this study suggested that COVID-19 in pregnancy was mainly occurs in the early and progressive stages, based on CT images. Chest CT scans showed small patchy or patchy GGO, distributed in the peripheral zone of both the lungs, with partial consolidation, a sign of intra-bronchial air-bronchogram and was accompanied by small bilateral pleural effusion. Thus, CT scan plays an important role in early screening of patients with atypical symptoms and/or negative nucleic acid testing and dynamic observation and efficacy evaluation of suspected or confirmed patients with COVID-19.

## Author Contributions

**Conceptualization:** ZhiYan Lu.

**Data curation:** XiaoMing Gong, Lu Song, Hang Li, ZhiYan Lu.

**Formal analysis:** Lu Song, ZhiYan Lu.

**Funding acquisition:** HengNing Ke, ZhiYan Lu.

**Methodology:** XiaoMing Gong, HengNing Ke.

**Project administration:** HengNing Ke, ZhiYan Lu.

**Resources:** XiaoMing Gong, Hongjun Li, ZhiYan Lu.

**Software:** HengNing Ke, ZhiYan Lu.

**Writing – original draft:** XiaoMing Gong.

**Writing – review & editing:** Hang Li, Li Li, Wei Jin, KaiHu Yu, Xiaochun Zhang, Hongjun Li, HengNing Ke, ZhiYan Lu.

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
