## [Decision Letter · Decision Letter 0]

15 Apr 2020

PONE-D-20-04574

CT characteristics and diagnostic value of COVID-19 in pregnancy

PLOS ONE

Dear Mr Lu,

Thank you for submitting your manuscript to PLOS ONE. After careful consideration, we feel that it has merit but does not fully meet PLOS ONE’s publication criteria as it currently stands. Therefore, we invite you to submit a revised version of the manuscript that addresses the points raised during the review process.

Please take into account the detailed comments listed below by the external reviewers when you prepare your Revision.

We would appreciate receiving your revised manuscript by May 30 2020 11:59PM. To enhance the reproducibility of your results, we recommend that if applicable you deposit your laboratory protocols in protocols.io, where a protocol can be assigned its own identifier (DOI) such that it can be cited independently in the future. For instructions see: http://journals.plos.org/plosone/s/submission-guidelines#loc-laboratory-protocols

We look forward to receiving your revised manuscript.

Kind regards,

Oliver Schildgen

Academic Editor

PLOS ONE

3. Please include your tables as part of your main manuscript and remove the individual files. Please note that supplementary tables (should remain/ be uploaded) as separate "supporting information" files

4. Thank you for including your ethics statement: 'The project has been reviewed by the Ethics Committee of Zhongnan Hospital of Wuhan University'.

a. Please amend your current ethics statement to confirm that your named institutional review board or ethics committee specifically approved this study.

5. Thank you for stating the following in your manuscript:

"National Nature Project(81572902)，Chinese Center for Disease Control and Disinfection Special Fund（ 201865） Hubei Provincial Health and Health Commission Project （Wj2017M044）；Science and Technology Innovation Cultivation Fund of Zhongnan Hospital of

Wuhan University（znpy2018033)"

"the sponsors or funders play an impormant role in the study design, data collection and analysis, decision to publish, or preparation of the manuscript"

Reviewers' comments:

Reviewer's Responses to Questions

**Comments to the Author**

1. Is the manuscript technically sound, and do the data support the conclusions?

Reviewer #1: Partly

2. Has the statistical analysis been performed appropriately and rigorously? 

Reviewer #1: N/A

3. Have the authors made all data underlying the findings in their manuscript fully available?

Reviewer #1: Yes

4. Is the manuscript presented in an intelligible fashion and written in standard English?

Reviewer #1: Yes

5. Review Comments to the Author

Reviewer #1: Review: “CT characteristics and diagnostic value of COVID-19 in pregnancy”

General comments:

This is a research article involving 10 pregnant women with COVID-19 in Wuhan from January 20, 2020 to February 6, 2020. Among them, 6 patients were laboratory- confirmed and 4 clinically-diagnosed cases. COVID-19 is a rapidly evolving pandemic. Pregnant woman is a special population. Limited data are available about COVID-19 during pregnancy. The understanding of clinical and radiological features of COVID-19 during pregnancy is important for the outcomes of mothers and fetuses.

This study retrospectively described the CT characteristics and assessed the diagnostic value for COVID-19 in pregnancy, which could help to interpret COVID-19 during pregnancy. I recommend a publication of this paper with major revisions.

Abstract:

1. Methods: “Clinical and chest CT data were collected and clinical symptoms, laboratory indicators, and CT images were analyzed to explore CT characteristics and diagnostic value of pregnancy with COVID-19”. The “…of pregnancy with COVID-19” should be written as“… for COVID-19 during pregnancy.”

Clinical symptoms and laboratory indicators were also analyzed. Were there any typical or atypical features need to be presented in Results?

2. Results: GGO: abbreviation should not be used for the first presentation.

3. Results: “Dynamic observation was performed in 4 cases after treatment”. Three cases showed progression or improvement, what was the follow-up time when progression or improvement occurred?

4. Conclusions: “The CT characteristics of COVID-19 in pregnancy are mainly in early and progressive stages”. What are the CT characteristics you are supposed to summarize for COVID-19 diagnosis?

Key words: OK

Introduction:

1. “Pregnant women are also susceptible population, and they are more likely to have complications than others, and even progress to severe cases.” Do you mean the pregnant women with COVID-19 are more likely to have complications or other concomitant pathogens? Do you have some references?

2. “However, COVID-19 in pregnancy is often found to be associated with pleural effusion.” Why? Do you have references? Is the pleural effusion possible to be intrinsically related with the pregnant status?

3. The introduction is too simple to state the current studies of COVID-19.

Materials and Methods:

1. 1 Patients

(1) “Patients an clinical and laboratory findings”. Please revise it.

(2) Did the Institutional Ethics Committee approve the study?

(3) “Among these 10 pregnant women, 6 tested positive for novel coronavirus nucleic acid”. How about the sample and test methods? Use throat swab samples and RT-PCR test?

(4) “Their age ranges from 26 to 40 year old, with an average age of 30.” The sample size was small, were the age, gestational age, and onset time in normal distribution? If not, median data were more appropriate.

1.2 Examination method

(1) Did all the pregnant preform CT scans before delivery?

(2) The thickness and interval was 1.0mm? How about the CT dose?

(3) Image interpretation can be described in another paragraph.

(4) Were there any low dose CT technologies implemented in the present study?

Results:

1. Distribution: Did you analyze the peripheral or non-peripheral distribution?

2. GGO: abbreviation should not be used for the first presentation.

3. “Other radiographic signs were also found that there were 6(60%) cases with small bilateral pleural effusions (figure 3) and no case with lymphadenopathy.” Need to be revised for avoiding the confusion. What were the diagnostic criteria for lymphadenopathy? Please add the criteria in “Materials and Methods” part.

Discussion:

1. Please state the main findings of your study in the first paragraph.

2. “6/10(60%) cases have small bilateral pleural effusion, which is not in line with reports that pleural effusion is rare according to Diagnosis and treatment of pneumonia for novel coronavirus infection (trial version 5)”. Please check whether the pleural effusion is associated with COVID-19 or pregnant status.

3. “all the 10 patients were in the early and progressive stages, and there was no critical stage; 6 cases (60%) were at early stage based on CT images: 2 cases (20%) showed single small patchy GGO in one lobe, and 4 cases (40%) showed multiple patchy GGO in the periphery of both lungs. Among them one case was normal in the first CT scan, which was consistent with the atypical COVID-19 performance reported by Chung M et al. [12]. The other 4 cases (40%) were at progressive stage, showing multiple small or large patchy GGO in both lungs. Some lesions were consolidated with sign of intra-bronchial air-bronchogram.” How to define the progressive stage? Peripheral distribution is common for COVID-19; however, the distribution was not demonstrated in the abstract or results.

4. Except for the pleural effusion, were there other CT characteristics for COVID-19 in pregnancy, which could help for the diagnosis?

5. Limitation part should be restructured before conclusion.

Please re-edit the language from abstract to discussion.

References: Please update some epub ahead of print references.

6. PLOS authors have the option to publish the peer review history of their article (what does this mean?). If published, this will include your full peer review and any attached files.

Reviewer #1: No

---

## [Author Response · Author response to Decision Letter 0]

20 May 2020

Abstract: 

1. Methods: “Clinical and chest CT data were collected and clinical symptoms, laboratory indicators, and CT images were analyzed to explore CT characteristics and diagnostic value of pregnancy with COVID-19”. The “…of pregnancy with COVID-19” should be written as“… for COVID-19 during pregnancy.”

Clinical symptoms and laboratory indicators were also analyzed. Were there any typical or atypical features need to be presented in Results?

[Response] : A correction has been made in the revised manuscript . The “…of pregnancy with COVID-19” had been written as“… for COVID-19 during pregnancy.Clinical symptoms and laboratory indicators were also analyzed. Laboratory examination showed that white blood cell account was normal in 9 cases, and slightly higher in 1 case (10.23×109). The lymphocyte ratio was decreased by 12% and 14% in 2 cases,. CRP was elevated in 7 cases, ranging from 21.16-60.3 mg/L. D-dimer was increased in 8 cases, ranging from 507-2141ng/ml. Total protein was low in 6 cases, ranging from 35.3-56.5mg/L.

2. Results: GGO: abbreviation should not be used for the first presentation.

[Response]: We regret not having explained the abbreviation more clearly. GGO, The full name, ground glass opacity, has been added in the revised manuscript.

3.Results: “Dynamic observation was performed in 4 cases after treatment”. Three cases showed progression or improvement, what was the follow-up time when progression or improvement occurred? 1 case showed progressed after treatment 7 days with lesion area expanding; and the other 2 cases showed improved after treatment 6 days or 12 days with the density of the lesion becoming lighter, the area of lesions decreasing and linear opacity appearing.

[Response]: 1 case showed progressed after 7 days of treatment with lesion area expanding; and the other 2 cases showed improved after 14 days of treatment with the density of the lesion becoming lighter, the area of lesions decreasing and linear opacity appearing. Changes were made in line20-25 in the abstract.

4. Conclusions: “The CT characteristics of COVID-19 in pregnancy are mainly in early and progressive stages”. What are the CT characteristics you are supposed to summarize for COVID-19 diagnosis?

[Response]: “The CT characteristics of COVID-19 in pregnancy are mainly in early and progressive stages”. The main manifestations are multiple pure ground glass density shadows under the pleura of the two lungs, and some solid changes. There are bronchial inflation signs, and no white lung signs. Changes were made in line 25-32 in the abstract.

Key words: OK

[Response]:Thank you for the favorable comments.

Introduction: 

1. “Pregnant women are also susceptible population, and they are more likely to have complications than others, and even progress to severe cases.” Do you mean the pregnant women with COVID-19 are more likely to have complications or other concomitant pathogens? Do you have some references?

[Response]: We regret not having explained these points more clearly. “Pregnant women are also susceptible population, and they are more likely to have complications than others, and even progress to severe cases.” We meant the pregnant women with COVID-19 are more likely to have complications . We have updated some newly references. The referrence is Liu H, Wang LL, Zhao SJ,et,al.Why are pregnant women susceptible to COVID-19? An immunological viewpoint. J Reprod Immunol. 2020 Mar 19;139:103122. DOI: 10.1016/j.jri.2020.103122. [Epub ahead of print]

2. “However, COVID-19 in pregnancy is often found to be associated with pleural effusion.” Why? Do you have references? Is the pleural effusion possible to be intrinsically related with the pregnant status?

[Response]: “However, COVID-19 in pregnancy is often found to be associated with pleural effusion.” This is a novel phenomenon we discovered during our work. There is no relevant literature report at present. As for the relationship between pleural effusion and pregnancy combined with new coronavirus pneumonia, it was described in detail in the discussion section. As for the mechanism and relevance, further research are neeeded..

3. The introduction is too simple to state the current studies of COVID-19.

[Response]: Thank you for your kind suggestion. .After repeatedly reading the literature and thinking, I have strengthened the description of the relevant knowledge of the COVID-19 in the revised manuscript. We added some new coronavirus incidence and epidemiological characteristics, Changes were made in line 2-8 and line 14-20 in the Introduction.

Materials and Methods:

1.1 Patients

(1) “Patients an clinical and laboratory findings”. Please revise it.

[Response]: “Patients an clinical and laboratory findings”has been replaced by “Study design and patients ”

(2) Did the Institutional Ethics Committee approve the study?

[Response]: Yes, This study was reviewed and approved by the Medical Ethical Committee of Zhongnan Hospital of Wuhan University (approval number 2020037). Written informed consent was obtained from each enrolled patient. 

(3) “Among these 10 pregnant women, 6 tested positive for novel coronavirus nucleic acid”. How about the sample and test methods? Use throat swab samples and RT-PCR test?

[Response]: Maternal throat swab samples were collected and tested for SARS-CoV-2 with the Chinese Center for Disease Control and Prevention (CDC) recommended RT-PCR Kit (BioGerm, Shanghai, China), following WHO guidelines for qRT-PCR.5–7 All samples were processed simultaneously at the Department of Clinical Laboratory of Zhongnan Hospital and State Key Laboratory of Virology/Institute of Medical Virology, School of Basic Medical Sciences, Wuhan University. Positive confirmatory cases of COVID-19 infection were defined as those with a positive test result from either laboratory.

(4) “Their age ranges from 26 to 40 year old, with an average age of 30.” The sample size was small, were the age, gestational age, and onset time in normal distribution? If not, median data were more appropriate. 

[Response]: “Their age ranges from 26 to 40 year old, with an average age of 30.” The sample size was small, were the age, gestational age, and onset time in abnormal distribution, median average is 27 year old.

1.2 Examination method

(1) Did all the pregnant preform CT scans before delivery? 

[Response]: All the ten pregnant preform CT scans before delivery.

(2) The thickness and interval was 1.0mm? How about the CT dose?

[Response]: pitch, =1.0; section thickness, =0.625mm; interval=5.0mm; axial reconstruction thickness=0.625mm; matrix 512*512 . All the ten pregnant preform CT scans before delivery, the CT dose was 411mGy.

(3) Image interpretation can be described in another paragraph.

[Response]: Image interpretation have been described in another paragraph in the revised manuscript. Changes were made in line 13 in the Examination method.

(4) Were there any low dose CT technologies implemented in the present study? 

[Response]:The low-dose dual-energy CT scan was not used. Due to the epidemic situation and the prevention of cross-infection, a special machine is used to scan patients with all suspicious new coronavirus pneumonia.

Results:

1. Distribution: Did you analyze the peripheral or non-peripheral distribution?

[Response]: We have analyzed the peripheral or non-peripheral distribution, 10（100%）patients with lesions distributed in the peripheral and 4(40%) patients with lesions distributed in the non peripheral. Changes were made in line 9-11 in the result.

2. GGO: abbreviation should not be used for the first presentation.

[Response]: The full English name has been marked. ground glass opacity(GGO).

3. “Other radiographic signs were also found that there were 6(60%) cases with small bilateral pleural effusions (figure 3) and no case with lymphadenopathy.” Need to be revised for avoiding the confusion. What were the diagnostic criteria for lymphadenopathy? Please add the criteria in “Materials and Methods” part.

[Response]: “Other radiographic signs were also found that there were 6(60%) cases with small bilateral pleural effusions (figure 3) and no case with lymphadenopathy.” Need to be revised for avoiding the confusion. Lymphadenopathy was defined as a lymph node >1 cm in short-axis diameter. The sentence was added in materials and methods, Changes were made in line 13 in the examination method.

Discussion:

1. Please state the main findings of your study in the first paragraph.

[Response]: The CT characteristics of COVID-19 in pregnancy are multiple new lesions at early stage，most of the original lesions expand the scope of the lesion during progressing stage, and there are consolidations of varying sizes and degrees within the lesion. The original ground glass lesions can also be fused or partially absorbed. 6/10 with small bilateral pleural effusion.The main findings of our study has been elaborated in the first paragraph.

2. “6/10(60%) cases have small bilateral pleural effusion, which is not in line with reports that pleural effusion is rare according to Diagnosis and treatment of pneumonia for novel coronavirus infection (trial version 5)”. Please check whether the pleural effusion is associated with COVID-19 or pregnant status.

[Response]: “6/10(60%) cases have small bilateral pleural effusion, which is not in line with reports that pleural effusion is rare according to Diagnosis and treatment of pneumonia for novel coronavirus infection (trial version 5)”. Authors analyzed the possible reasons as follows: a) the total protein of these 6 patients was decreased to different degrees, ranging from 35.3mg to 56.5mg/L, and a small amount of pleural effusion might be caused by the decrease of plasma colloid osmotic pressure due to slight hypoproteinemia; b) CRP was elevated in all the 6 patients, ranging from 21.17.85-60.3mg/L, showing certain inflammatory reactions. Infection made the visceral pleura thickened and congested, and increased vascular permeability induced pleural effusion. Exudate moved across the visceral pleura in a gradient, causing pleural effusion [6]. c) all the cases in this group were women with late pregnancy at 36-38 weeks of gestation, and late pregnant women were prone to chest leakage due to increased blood volume. As for the mechanism and relevance, it needs further research by researchers.

3. “all the 10 patients were in the early and progressive stages, and there was no critical stage; 6 cases (60%) were at early stage based on CT images: 2 cases (20%) showed single small patchy GGO in one lobe, and 4 cases (40%) showed multiple patchy GGO in the periphery of both lungs. Among them one case was normal in the first CT scan, which was consistent with the atypical COVID-19 performance reported by Chung M et al. [12]. The other 4 cases (40%) were at progressive stage, showing multiple small or large patchy GGO in both lungs. Some lesions were consolidated with sign of intra-bronchial air-bronchogram.” How to define the progressive stage? Peripheral distribution is common for COVID-19; however, the distribution was not demonstrated in the abstract or results.

[Response]: As the progressive stage, multiple new lesions are common, and the CT appearance of the new lesion is similar to that of the early lesions. Most of the original lesions expand the scope of the lesion, and there are consolidations of varying sizes and degrees within the lesion. The original ground glass lesions can also be fused or partially absorbed. After fusion, the scope and shape of the lesion often change, not completely distributed along the bronchial vascular bundle.(From:The Radiological Branch of the Chinese Medical Association. The Radiological Diagnosis of New Coronavirus Pneumonia: Recommendations from Experts of the Radiological Branch of the Chinese Medical Association (First Edition). Chinese Journal of Radiology. 2020, 54: E001-E001).Peripheral distribution is common for COVID-19; and the distribution was already shown in the abstract and results .Changes were made in line16-17)in the anstract.

4. Except for the pleural effusion, were there other CT characteristics for COVID-19 in pregnancy, which could help for the diagnosis?

[Response]: Except for the pleural effusion, were there other CT characteristics for COVID-19 in pregnancy, The lesions were mild, the lesions were not very extensive, and there were no cases of white lungs. The typical signs of paving stones in both lungs were also rare.This characteristics could help for the diagnosis.

5. Limitation part should be restructured before conclusion.

[Response]: Limitation part had been restructured before conclusion.

Please re-edit the language from abstract to discussion.

[Response]: The language of the revised manuscript has now been checked by a native English speaker.

References: Please update some epub ahead of print references.

[Response]:We have updated some epub ahead of print references.

[8] Chung M, Bernheim A, Mei X，et,al.CT Imaging Features of 2019 Novel Coronavirus (2019-nCoV).[Epub ahead of print].Radiology. 2020 Feb 4:200230. doi: 10.1148/radiol.2020200230. Chung M, Bernheim A, Mei X, et al. CT Imaging Features of 2019 Novel Coronavirus (2019-nCoV). Radiology. 2020;295(1):202–207. doi:10.1148/radiol.2020200230

[9] Song F, Shi N, Shan F,et,al.Emerging Coronavirus 2019-nCoV Pneumonia.[Epub ahead

of print].Radiology. 2020 Feb 6:200274. doi: 10.1148/radiol.2020200274.Song F, Shi N, Shan F, et al. Emerging 2019 Novel Coronavirus (2019-nCoV) Pneumonia. Radiology. 2020;295(1):210–217. doi:10.1148/radiol.2020200274

[10] Wang D, Hu B, Hu C,et al.Clinical Characteristics of 138 Hospitalized Patients With 2019 Novel Coronavirus-Infected Pneumonia in [J].JAMA. 2020 Feb 7.[Epub ahead of print].DOI:10.1001/jama.2020.1585. Wang D, Hu B, Hu C, et al. Clinical Characteristics of 138 Hospitalized Patients With 2019 Novel Coronavirus-Infected Pneumonia in Wuhan, China [published online ahead of print, 2020 Feb 7]. JAMA. 2020;e201585. doi:10.1001/jama.2020.1585

---

## [Editor Report · Decision Letter 1]

10 Jun 2020

CT characteristics and diagnostic value of COVID-19 in pregnancy

PONE-D-20-04574R1

Dear Dr. Lu,

We’re pleased to inform you that your manuscript has been judged scientifically suitable for publication and will be formally accepted for publication once it meets all outstanding technical requirements.

Kind regards,

Oliver Schildgen

Academic Editor

PLOS ONE
---

## [Editor Report · Acceptance letter]

16 Jun 2020

PONE-D-20-04574R1 

CT characteristics and diagnostic value of COVID-19 in pregnancy 

Dear Dr. Lu:

I'm pleased to inform you that your manuscript has been deemed suitable for publication in PLOS ONE. Congratulations! Your manuscript is now with our production department. 

Kind regards, 

on behalf of

Prof. Oliver Schildgen 

Academic Editor

PLOS ONE